# Estimation of Genetic Parameters for Conformation Traits and Milk Production Traits in Chinese Holsteins

**DOI:** 10.3390/ani13010100

**Published:** 2022-12-27

**Authors:** Xiaoshu Xue, Honghong Hu, Junxing Zhang, Yanfen Ma, Liyun Han, Feng Hao, Yu Jiang, Yun Ma

**Affiliations:** 1Key Laboratory of Ruminant Molecular Cell Breeding, Ningxia Hui Autonomous Region, School of Agriculture, Ningxia University, Yinchuan 750021, China; 2Ningxia Farming Cultivate Helanshan Dairy Co., Ltd., Yinchuan 750021, China; 3College of Animal Science and Technology, Northwest A&F University, Xianyang 712100, China

**Keywords:** Holstein dairy cows, conformation traits, milk production traits, genetic parameters

## Abstract

**Simple Summary:**

Genetic parameters can reflect some regulars of quantitative traits in the population, and the estimation of genetic parameters is very important for cow breeding. The conformation traits of dairy cows are closely related to their milk production and reproduction, which are among the most important traits in dairy breeding. At the same time, milk production of dairy cows has always been the focus of attention at the cattle farm. Studies have shown that there are genetic correlations between conformation traits and milk production traits. For example, taller cows produce more milk. The aim of this study was to investigate the genetic correlations between conformation traits and milk production traits in Chinese Holstein dairy cows. In this study, we estimated the heritability of 23 conformation traits and five milk production traits and the genetic correlation between them using DMU software combined with single-trait and multiple-trait animal models. The heritability of conformation traits was medium in Holstein cows in Ningxia, China. In addition, there were positive genetic correlations between most conformation traits and milk production traits.

**Abstract:**

The objective of this study was to explore the genetic parameters of conformation traits and milk production traits in Chinese Holstein cattle and to provide a reference for dairy cattle breeding. We collected the phenotypic data of 23 conformation traits and five milk production traits of Chinese Holsteins and used animal models to estimate the genetic parameters of conformation traits and milk production traits. The estimated heritability of conformation traits ranged from 0.11 (angularity) to 0.37 (heel depth) and the genetic correlation between conformation traits ranged from −0.73 (bone quality and rear leg-rear view) to 0.76 (chest width and loin strength). The heritability of milk production traits ranged from 0.23 (somatic cell score) to 0.50 (305-d milk yield). The estimated values of genetic correlation between conformation traits and milk production traits ranged from −0.56 (heel depth and 305-d milk yield) to 0.57 (udder texture and milk fat percentage). There was a positive genetic correlation between most conformation traits and milk fat percentage, but a weak negative genetic correlation with milk yield. Strengthening the moderately and highly heritable milk production and conformation traits, especially the selection of rear udder traits and body shape total score, will be beneficial in improving the performance of dairy cows.

## 1. Introduction

Conformation traits are closely related to production, reproduction, mastitis resistance, and herd life of dairy cows [1,2,3,4,5]. Conformation traits have been used in dairy breeding programs in many countries since the 1990s. Scientists can accurately and scientifically evaluate the heritability of each character estimate breeding value through conformation traits, and directly reflect the production performance and health level of dairy cows, thus indirectly reflecting their production capacity. Therefore, it is of great significance to study conformation traits for genetic evaluation and improvement of target traits [6]. Milk production is one of the most important product characteristics of dairy cows and also the main target character of dairy cows breeding in various countries. Many scholars have reported on the genetic analysis of cows’ conformation traits and milk production traits [7,8,9,10,11,12,13,14]. For example, Short et al. [14] reported a genetic correlation of 0.06 for the final score of conformation traits and milk yield. Sartori et al. [15] reported that breast volume was positively correlated with milk yield, with an average correlation of 0.427. Misztal et al. [16] reported that there was a negative genetic correlation between udder depth and milk yield. Monardes et al. [17] and Welper et al. [18] found that mammary system traits are also correlated with somatic cell score. Kelm et al. [19] showed that cows with deeper and wider udders have significantly greater milk production. In conclusion, conformation traits are correlated with many milk production traits, and some conformation traits can indirectly affect the health and milk production performance of dairy cows. The Ningxia Hui Autonomous Region, as a relatively well-developed region for the dairy industry in China, has large-scale farms and a good herd management system that plays a key supporting role in research involving Chinese Holsteins. This study was based on the DHI (Dairy Herd Improvement) records and first parity identification data of Chinese Holstein cattle from the Ningxia Hui Autonomous Region. We used the DMU computer package combined with single-trait and multiple-trait animal models to estimate the heritabilities of conformation and milk production traits, as well as the correlations between the traits. The objective was to provide a reference for improving the production performance and breeding value of dairy cows by estimating the genetic parameters of conformation traits and milk production traits in Chinese Holstein cattle.

## 2. Materials and Methods

### 2.1. Data Sources and Collation

#### 2.1.1. Data Sources

The data of this study were collected at the Ningxia Farming Cultivate Helanshan Dairy Co., Ltd. We collected data on the conformation traits and milk yield of 64,972 Chinese Holstein dairy cows from 74 farms in the Ningxia Hui Autonomous Region from 2007 to 2020. The five milk production traits studied were milk yield, milk fat percentage, milk protein percentage, somatic cell score, and 305-d milk yield. Conformation traits included 23 linear traits: stature, height at the front end, trunk size, chest width, body depth, loin strength, pin setting, pin width, foot angle, heel depth, bone quality, set of rear legs, rear leg-rear view, udder depth, udder texture, median suspensory, fore attachment, fore teat placement, fore teat length, rear attachment height, rear attachment width, rear teat placement, and angularity. Cows’ conformation traits were recorded using a 9-point scale to evaluate linear character, including frame capacity, rump, feet and legs, mammary system, and dairy character. The assessors determined the conformation trait scores 30~180 days after the first calving. SAS V9.0 was used to trace the pedigrees to three generations and Excel 2019 was used to organize the pedigrees. A total of 2974 bull and 81,256 cow pedigrees were traced. Trait names, abbreviations, and definitions are presented in Table 1.

Values for the conformation and milk production traits were determined by a professional assessor. DHI data is determined in strict accordance with national standards. The measurement standards of conformation traits and milk production traits were based on the Code of Practice of Type Classification in Chinese Holstein and Technical of Chinese Holstein Cattle Performance Test in Dairy Data Center of China (https://www.holstein.org.cn/newsIndex.jsp (accessed on 15 July 2022)).

#### 2.1.2. Data Collation

According to the industry standard Chinese Holstein Cattle Performance Measurement Technical Specification [20], an FP > 7.0% or <2.0% and a PP > 5.0% or <2.0% is abnormal data, and we deleted values outside this range above. Data with 305MY, FP, and PP phenotypic values outside ±3 SD from the mean were also deleted. The final screening criteria for milk production data: (1) The range of MY is 50–80 kg per day; (2) The range of 305MY is 2000–16,000 kg; (3) The range of FP is 2.0–7.0%; (4) The range of PP is 2.0–5.0%. Since the data distribution of somatic cell numbers is not normal, it often needs to be transformed into SCS in statistical analysis. The widely used conversion formula is proposed by Dabdoub et al. [21], and is as follows:*SCS* = *log**_2_*(*SCC*/100000) + 3

The quality control standard of conformation traits identification states: (1) All herds measuring fewer than 100 cattle were eliminated; (2) The offspring of bulls with fewer than 50 daughters were deleted; (3) Deleted offspring of bulls with fewer than 7 daughter distribution farms. A total of 11,019 conformation observations and 17,462 DHI observations were selected for heritability analysis. Subsequently, 7923 first parity cows that met the criteria were selected as the analysis objects for further statistical analysis of the correlation between conformation traits and milk production traits. After quality control, the cattle from 18 farms in the Ningxia Hui Autonomous Region were selected. The cattle were from 18 farms in the Ningxia Hui Autonomous Region. The farms were of different sizes and had different feeding and management conditions. Because of this, we included farm as a fixed effect in the statistical model.

### 2.2. Statistical Analysis Methods

Data were analyzed with a linear animal model using the Average Information Restricted Maximum Likelihood (AI-REML) algorithm in the DMU package [22]. The DMUAI module in DMU V6 R5.2 was used to estimate heritabilities and genetic correlations with single-trait and multiple-trait animal models, respectively. AIREML was combined with the Expectation Maximization (EM) algorithm to estimate the components of variance, and finally, the genetic parameters were estimated. The fixed effects of conformation traits included farm, year-seasons of measurement, and assessor. The fixed effects of milk production traits included farm, year-seasons of calving, and year-seasons of measurement. Because management levels and technology were different, each farm was treated as a level of the fixed effect. Calving years and measurement years were divided into 13 levels (from 2007 to 2020, one level per year). The seasons were divided into four levels (spring: March to May; summer: June to August; autumn: September to November; winter: December to February). Each assessor acted as a level.

Analysis model of conformation traits:*y_ijkl_* = *μ* + *farm_i_* + *tys_j_* + *i_k_* + *id_l_* + *e_ijkl_*

Analysis model of milk production traits:*y_ijml_* = *μ* + *farm_i_* + *tys_j_* + *bys_m_* + *id_l_* + *e_ijml_*
where, *y_ijkl_* was the observed value of conformation traits; *y_ijml_* was the observed value of milk production traits; *μ* was the population mean of all observed values; *farm_i_* was the fixed effect of the i-th farm; *tys_j_* was the fixed effect of the *j*-th year-season of measurement; *i_k_* is the fixed effect of the *k*-th assessor; *bys_m_* was the fixed effect of the *k*-th year-season of calving; *id_l_* was the individual additive genetic effect vector; and *e_ijkl_* and *e_ijml_* were random residual effect vectors.

Heritability calculation formula:$$h^{2}=\frac{V_{A}}{V_{A}+V_{E}}\;$$

Genetic correlation calculation formula:$$r_{A}=\frac{Cov\left(a_{1},a_{2}\right)}{\sqrt{\sigma_{a1}^{2}\sigma_{a2}^{2}}}\;$$

Phenotypic correlation calculation formula:$$r_{P}=\frac{Cov\left(p_{1},p_{2}\right)}{\sqrt{\sigma_{p1}^{2}\sigma_{p2}^{2}}}\;$$
where, *h*^2^ = heritability, *V_A_* = additive genetic variance, and *V_E_* = environmental variance. *r_A_* is the genetic correlation of traits, *Cov*(*a*_1_,*a*_2_) is the additive genetic covariance of trait 1 and trait 2, and *σ_a_*_1_^2^ and *σ_a_*_2_^2^ represent the additive genetic variances of trait 1 and trait 2. *r_P_* is the phenotypic correlation between traits, *Cov*(*p*_1_,*p*_2_) is the phenotypic covariance of trait 1 and trait 2, and *σ_p_*_1_^2^, *σ_p_*_2_^2^ is the phenotypic variances of each trait.

## 3. Results

### 3.1. Descriptive Statistics of Conformation Traits and Production Traits

The means of 23 linear scores for conformation traits ranged from 4.97 (UD) to 6.85 (ST). The standard deviation ranged from 0.72 (BD) to 1.38 (UD). The optimal gap from the ideal score ranged from −0.21 (FTP) to 3.85 (RAH). The coefficients of variation were all less than 30% (Table 2).

Descriptive statistics for milk production traits in Chinese Holsteins are shown in Table 3. The coefficient of variation was less than 30% for all traits except SCS (55.43%) (Table 3).

### 3.2. Heritability Estimates for Conformation Traits and Milk Production Traits

Heritabilities of conformation traits ranged from 0.11 (ANG) to 0.37 (HD) (Table 4). ST, CW, LS, PW, SORL, RLRV, UD, MSL, FA, FTL, FAN, HD, BQ, RAH, RAW and RTP (0.2 < *h*^2^ < 0.5) were moderately heritable traits.

Heritabilities of milk production traits ranged from 0.23 (SCS) to 0.50 (305MY) (Table 5). Except for 305MY (0.50) which was highly heritable (*h*^2^ ≥ 0.5), all the other milk production traits were moderately heritable (0.2 < *h*^2^ < 0.5).

### 3.3. Correlations between Conformation Traits

The genetic correlations between conformation traits in Chinese Holstein cattle ranged from −0.75 (UD and ANG) to 0.95 (ST and TS). ST and TS (0.95), HFE and TS (0.54), HFE and PW (0.93), HFE and FTL (0.69), TS and PW (0.65), TS and FTL (0.72), TS and ANG (0.69), CW and LS (0.76), CW and PW (0.60), LS and PS (0.62), LS and UT (0.88), UT and FTL (0.64), UT and ANG (0.88), MSL and FTP (0.67), and MSL and RTP (0.69) showed high correlations (shown in bold). Phenotypic correlations between conformation traits ranged from −0.52 (UD and ANG) to 0.68 (ST and TS) (Figure 1 and Figure 2). Detailed information can be found in the Appendix A (Appendix A and Appendix A).

The genetic correlations among frame capacity traits ranged from 0.15 to 0.76. The genetic correlation between BD and CW was low (0.15), but the genetic correlation between CW and LS was large and positive (0.76), indicating a moderately positive genetic correlation. The phenotypic correlations among frame capacity traits ranged from 0.13 (BD and LS) to 0.24 (ST and BD, ST and CW), and all the phenotypic correlations were moderate to weak in magnitude. Both genetic and phenotypic correlations were positive between PS and PW, but the phenotypic correlation was small (0.08).

The genetic correlations between feet and leg traits ranged from −0.73 to 0.58, in which there were negative genetic correlations between BQ and HD (−0.56), BQ and SORL (−0.73), FAN and SORL (−0.22), HD and RLRV (−0.21), and SORL and RLRV (−0.41). The other traits showed positive correlations. In addition, phenotypic correlations between feet and leg traits ranged from −0.16 to 0.20, with negative correlations including BQ and FAN (−0.02), BQ and SORL (−0.12), FAN and SORL (-0.09), and SORL and RLRV (−0.16). Phenotypic correlations among other traits were positive but weak.

The genetic correlations between mammary system traits ranged from −0.57 to 0.69. The genetic correlations between MSL and FTL (−0.09), FTP and FTL (−0.57), FTL and RAH (−0.03), UD and RAW (−0.24), RAH and RAW (−0.22), and FTL and RTP (−0.12) were negative. The genetic correlations were positive for the remaining trait combinations. The phenotypic correlations among the selected traits ranged from −0.13 to 0.26, with weak positive correlations among the traits except between UD and RAW (−0.13).

### 3.4. Correlations between Conformation and Milk Production Traits

The genetic correlations between MY, FP, PP, SCS, 305MY, and conformation traits ranged from −0.31 (PS and MY) to 0.34 (ST and MY), −0.33 (PS and FP) to 0.57 (UT and FP), −0.32 (PS and PP) to 0.38 (UT and PP), −0.43 (HD and SCS) to 0.34 (BD and SCS), and −0.56 (FAN and 305MY) to 0.32 (BD and 305MY) (Table 6). The phenotypic correlations of conformation traits and milk production traits ranged from −0.13 (HD and MY) to 0.14 (TS and MY), −0.12 (HD and FP) to 0.24 (UT and FP), −0.13 (HD and PP) to 0.10 (UT and PP), −0.13 (HD and SCS)~0.09 (BQ and SCS), and −0.15 (HD and 305MY)~0.33 (UT and 305MY) (Table 6). The phenotypic correlations between most conformation traits and milk production traits were less than 0.10.

## 4. Discussion

### 4.1. Heritabilities of Conformation Traits and Milk Production Traits

Heritabilities of conformation traits ranged from 0.11 (ANG) to 0.37 (HD). The results were similar to those of Kadarmideen et al. [23] (0.08~0.46), Dal Zotto et al. [24] (0.07~0.32), and the latest results of Canadian Holstein cattle [25] (0.04~0.47). Heritability of FTP (0.14) was close to that reported by Dadpasand et al. [26] for Iranian Holstein cattle (0.13). However, it is significantly lower than the results of Van der Laak et al. [27] (0.35) and Zink et al. [28] (0.39). The results showed that LS (0.32), BQ (0.37), HD (0.37), and RTP (0.33) were moderately heritable (0.2 < *h*^2^ < 0.5). Heritabilities of the other traits were moderate, except for 305MY (0.50), which may be mainly due to conformational traits. The moderate heritabilities may be due to the fact that conformation traits can be directly observed at an early age and are relatively simple to record [6]. In addition, the level of assessors and the amount of data were other reasons for the difference in the heritability of traits. Different assessors had different identification levels, and the identification data might be biased to some extent.

### 4.2. Genetic and Phenotypic Correlations among Conformation Traits

The range of phenotypic correlations between conformation traits was −0.52 (UD and ANG) to 0.68 (ST and TS) and the range of genetic correlations were −0.75 (UD and ANG) to 0.95 (ST and TS). The phenotypic correlation of PS and PW (0.08) was close to that of Oliveira et al. [25] (0.05).

There were weak phenotypic correlations among all the feet and leg traits and the genetic correlation of FAN and SORL (−0.22) was close to that of Roveglia et al. [29] (−0.27). In addition, there was a high genetic correlation between HD and RLRV (0.58).

Of the mammary system traits, the genetic correlation between FTP and RTP (0.49) was lower than that reported by Oliveira et al. [25] in Canadian Holstein cattle (0.63) and Bohlouli et al. [7] in Iranian Holstein cattle (0.60). The genetic correlation between UD and FTP (0.04) was significantly smaller than that of Tapki et al. [12] (0.46). Phenotypic correlations among mammary system traits were low, which finding agreed with the results of Bohlouli et al. [7].

### 4.3. Correlations between Conformation Traits and Milk Production Traits

ST, BD, and BQ in Chinese Holsteins had strongly positive genetic correlations with production traits. This indicated that cows with larger TS, deeper BD, and better bone quality had more milk, their milk contained more fat and protein, and it had a better somatic cell score, which finding is similar to the results of Bohlouli et al. [7]. It further indicated that cows with larger frame capacity had better milk production performance and higher milk yield. There were moderately negative genetic correlations between FAN and 305MY (−0.56) and RAW and 305MY (−0.43), while there were moderately negative genetic correlations between FAN and HD and 305MY (−0.34). Meanwhile, improving milk yield may lead to reduced heel depth and an increase in the risk of infection and limb and hoof diseases. Among the mammary system traits, UD was negatively correlated with 305MY (−0.12), FP (−0.12), PP (−0.12), and SCS (−0.17). This indicates that cows with deep udders have relatively poor milk traits and production. FTP was positively correlated with 305MY (0.07), close to the value reported by Ismael et al. (0.091) [30] for Serbian Holstein-Frisian cows. However, Oliveira et al. [25] reported that the correlation between UD and 305MY in Canadian Holsteins was −0.45 and that there were negative genetic correlations between RAW and FP and between RAW and PP, the same as in our study. The contrasting results may indicate that the data identified by different assessors were biased, or it may be caused by the difference in feeding management that leads to the difference in the linear scores of the measured cattle.

There were moderately strong genetic correlations of SCS with BD, PS, and HD (0.34, −0.39, and −0.43, respectively), indicating that frame capacity, rump, and feet and leg traits also affected somatic cell score and therefore affected the occurrence of mastitis. There was little correlation between HFE and SCS (−0.02), which was similar to a previous report [31]. Too long or too short nipple length would increase the somatic cell score [32]. To sum up, it is important to select for proper udder characteristics to improve milk production performance and ensure the udder health of cows.

Although conformation traits do not produce direct economic benefits, some conformation traits can affect the milk production performance of dairy cows through direct or indirect effects and can be used to identify and breed high quality, high yield, healthy, and long-lived dairy herds, which would improve the economic standing of dairy farms. Therefore, strengthening the selection of moderately to highly heritable conformation traits and milk production traits is beneficial to the production performance of dairy cows.

## 5. Conclusions

Among the conformation traits of Chinese Holstein cattle in Ningxia, LS (0.32), BQ (0.37), HD (0.37), and RTP (0.33) were medium heritability traits. Among the milk producing traits, 305MY (0.50) was highly heritable. These moderately heritable traits can be improved through systematic breeding and selection. Compared with other studies, the genetic correlation of the conformation traits of feet and leg traits has some deviation. The correlations between mammary system traits and milk production traits are very important. In this study, UD showed moderately negative genetic correlations with milk production traits, while both FTP (0.07) and RAW (−0.43) showed low genetic correlations with 305MY. There were also negative genetic correlations of UD (−0.17) and FTL (−0.02) with SCS.

## Figures and Tables

**Figure 1 animals-13-00100-f001:**
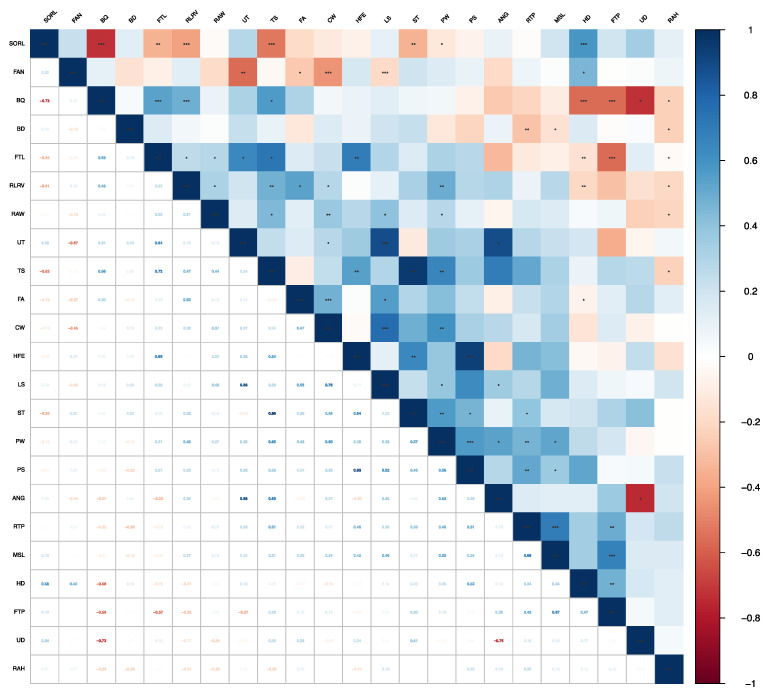
Genetic correlations between conformation traits in Chinese Holsteins. The upper triangle is the result of the significance test, and the lower triangle is the Pearson correlation coefficient. Blue represents a positive correlation and red represents a negative correlation. *: significant level less than 0.05, **: significant level less than 0.01, ***: significant level less than 0.001.

**Figure 2 animals-13-00100-f002:**
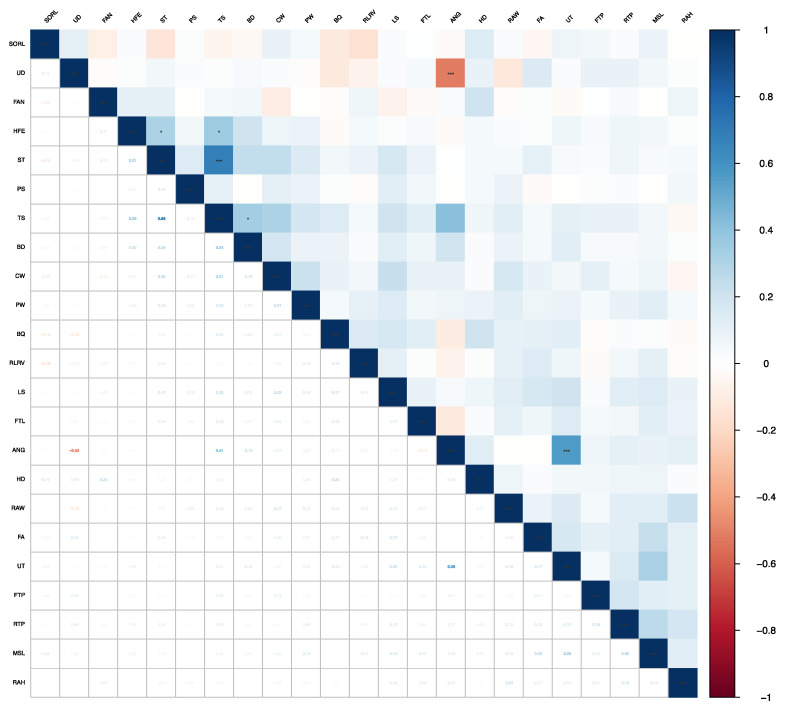
Phenotypic correlations between conformation traits in Chinese Holsteins. The upper triangle is the result of the significance test, and the lower triangle is the Pearson correlation coefficient. Blue represents a positive correlation and red represents a negative correlation. *: significant level less than 0.05, ***: significant level less than 0.001.

**Table 1 animals-13-00100-t001:** Abbreviations and trait names.

Abbreviations		Trait Name
Milk production traits	MY	dairy milk yield
	FP	milk fat percentage
	PP	milk protein percentage
	SCS	somatic cell score
	305MY	305-d milk yield
Frame capacity	ST	stature
	HFE	height at the front end
	TS	trunk size
	CW	chest width
	BD	body depth
	LS	loin strength
Rump	PS	pin setting
	PW	pin width
Feet and legs	FAN	foot angle
	HD	heel depth
	BQ	bone quality
	SORL	set of rear legs
	RLRV	rear leg-rear view
Mammary system	UD	udder depth
	UT	udder texture
	MSL	median suspensory
	FA	fore attachment
	FTP	fore teat placement
	FTL	fore teat length
	RAH	rear attachment height
	RAW	rear attachment width
	RTP	rear teat placement
Dairy character	ANG	angularity

**Table 2 animals-13-00100-t002:** Descriptive statistics for conformation traits in Chinese Holsteins ^1^.

Traits		Number of Records	Mean	SD	CV, %	Ideal Scores	Optimal Gap
Frame capacity	ST	10,914	6.85	1.19	17.37	8	1.15
	HFE	4539	5.67	1.02	17.99	7	1.33
	TS	4539	6.57	1.03	15.68	9	2.43
	CW	10,914	6.11	1.13	18.49	9	2.89
	BD	10,914	6.56	0.72	10.98	7	0.44
	LS	10,914	6.15	1.22	19.84	9	2.85
Rump	PS	10,914	5.07	1.04	20.51	5	−0.07
	PW	10,914	5.67	0.93	16.40	9	3.33
Feet and legs	FAN	10,914	5.30	1.08	20.38	7	1.70
	HD	10,914	5.37	1.03	19.18	9	3.63
	BQ	10,914	6.42	1.05	16.36	9	2.58
	SORL	10,914	5.08	1.20	23.62	5	−0.08
	RLRV	10,914	5.63	1.28	22.74	9	3.37
Mammary system	UD	10,248	4.97	1.38	27.77	5	0.03
	UT	5205	5.77	1.37	23.75	9	3.23
	MSL	10,914	5.21	1.33	25.53	9	3.79
	FA	10,914	5.33	1.36	25.52	9	3.67
	FTP	10,914	5.39	0.81	15.03	6	0.61
	FTL	10,914	5.21	0.88	16.89	5	−0.21
	RAH	10,914	5.15	1.30	25.24	9	3.85
	RAW	10,914	5.51	1.20	21.78	9	3.49
	RTP	10,914	5.90	0.95	16.10	6	0.10
Dairy character	ANG	9626	6.49	0.86	13.25	9	2.51

^1^ ST, stature; HFE, height at the front end; TS, trunk size; CW, chest width; BD, body depth; LS, loin strength; PS, pin setting; PW, pin width; FAN, foot angle; HD, heel depth; BQ, bone quality; SORL, set of rear legs; RLRV, rear leg-rear view; UD, udder depth; UT, udder texture; MSL, median suspensory; FA, fore attachment; FTP, fore teat placement; FTL, fore teat length; RAH, rear attachment height; RAW, rear attachment width; RTP, rear teat placement; ANG, angularity; SD, Standard Deviation; CV, Coefficient of Variation; ideal score: the most desirable score for each conformation trait; optimal gap: the difference between the mean and the ideal score for each conformation trait.

**Table 3 animals-13-00100-t003:** Descriptive statistics for milk production traits in Chinese Holsteins ^1^.

Traits	Number of Records	Mean	SD	CV, %	Min	Max
MY, kg	17,429	31.28	8.67	27.72	5.00	79.00
FP, %	17,429	3.84	0.65	16.93	2.00	6.98
PP, %	17,429	3.35	0.30	8.96	2.33	4.97
SCS	17,429	2.76	1.53	55.43	0.00	9.00
305MY, kg	17,429	9047.61	1963.59	21.70	2037.23	15,943.37

^1^ MY, milk yield; FP, milk fat percentage; PP, milk protein percentage; SCS, somatic cell score; 305MY, 305-d milk yield; SD, standard deviation; CV, coefficient of variation; Min, minimum; Max, maximum.

**Table 4 animals-13-00100-t004:** The heritability of conformation traits ^1^ in Chinese Holsteins.

Traits		*h* ^2^	SE	*σ_a_* ^2^	*σ_P_* ^2^	*σ_e_* ^2^
Frame capacity	ST	0.30	0.04	0.30	1.01	0.71
	HFE	0.11	0.04	0.08	0.78	0.70
	TS	0.19	0.05	0.15	0.79	0.64
	CW	0.24	0.04	0.22	0.91	0.69
	BD	0.12	0.02	0.06	0.48	0.42
	LS	0.32	0.04	0.32	1.02	0.69
Rump	PS	0.18	0.03	0.18	0.97	0.79
	PW	0.28	0.04	0.15	0.55	0.39
Feet and legs	FAN	0.14	0.03	0.12	0.81	0.69
	HD	0.37	0.05	0.21	0.57	0.36
	BQ	0.37	0.04	0.27	0.73	0.46
	SORL	0.30	0.04	0.30	1.01	0.71
	RLRV	0.27	0.04	0.36	1.35	0.99
Mammary system	UD	0.21	0.03	0.24	1.17	0.93
	UT	0.12	0.03	0.14	1.15	1.01
	MSL	0.28	0.04	0.37	1.32	0.95
	FA	0.25	0.04	0.30	1.21	0.90
	FTP	0.14	0.03	0.06	0.44	0.38
	FTL	0.28	0.04	0.20	0.72	0.52
	RAH	0.21	0.03	0.25	1.22	0.97
	RAW	0.20	0.03	0.19	0.92	0.73
	RTP	0.33	0.04	0.26	0.79	0.53
Dairy character	ANG	0.11	0.02	5440.90	51,525.05	46,084.15

^1^ ST, stature; HFE, height at the front end; TS, trunk size; CW, chest width; BD, body depth; LS, loin strength; PS, pin setting; PW, pin width; FAN, foot angle; HD, heel depth; BQ, bone quality; SORL, set of rear legs; RLRV, rear leg-rear view; UD, udder depth; UT, udder texture; MSL, median suspensory; FA, fore attachment; FTP, fore teat placement; FTL, fore teat length; RAH, rear attachment height; RAW, rear attachment width; RTP, rear teat placement; ANG, angularity; *h*^2^, heritability; SE, standard error; *σ_a_*^2^, additive genetic variance; *σ_P_*^2^, phenotypic varianc; *σ_E_*^2^, environmental variance.

**Table 5 animals-13-00100-t005:** The heritability of milk production traits ^1^ in Chinese Holsteins.

Traits	*h* ^2^	SE	σ_a_^2^	σ_P_^2^	σ_e_^2^
MY	0.47	0.05	26.04	54.95	28.91
FP	0.45	0.04	0.17	0.37	0.20
PP	0.30	0.03	0.02	0.07	0.05
SCS	0.23	0.03	0.43	1.87	1.44
305MY	0.50	0.04	1,446,520.69	2,911,383.06	1,464,862.37

^1^ MY, milk yield; FP, milk fat percentage; PP, milk protein percentage; SCS, somatic cell score; 305MY, 305-d milk yield; *h*^2^, heritability; SE, standard error; *σ_a_*^2^, additive genetic variance; *σ_P_*^2^, phenotypic varianc; *σ_E_*^2^, environmental variance.

**Table 6 animals-13-00100-t006:** Genetic and phenotypic correlations between conformation traits and milk production traits in Chinese Holsteins ^1^.

Traits	MY		FP		PP		SCS		305dMY	
	Genetic Correlation	Phenotypic Correlation	Genetic Correlation	Phenotypic Correlation	Genetic Correlation	Phenotypic Correlation	Genetic Correlation	Phenotypic Correlation	Genetic Correlation	Phenotypic Correlation
ST	0.34	0.08	0.30	0.07	0.32	0.08	0.23	0.08	0.22	0.07
HFE	−0.09	−0.02	−0.08	−0.02	0.03	−0.01	−0.07	−0.01	0.24	0.10
TS	0.32	0.14	0.34	0.15	0.19	0.05	0.05	0.02	−0.05	−0.03
CW	0.09	0.04	0.08	0.04	0.08	0.04	0.05	0.03	0.10	0.06
BD	0.32	0.08	0.31	0.08	0.31	0.08	0.34	0.08	0.32	0.11
LS	−0.06	−0.01	−0.06	−0.01	−0.07	−0.01	−0.05	0.01	−0.21	−0.03
PS	−0.31	−0.08	−0.33	−0.08	−0.32	−0.08	−0.39	−0.06	−0.07	−0.05
PW	0.05	−0.04	0.05	−0.03	0.04	−0.04	−0.06	−0.05	0.09	0.01
FAN	−0.15	−0.06	−0.15	−0.06	−0.15	−0.06	−0.27	−0.07	−0.56	−0.11
HD	−0.27	−0.13	−0.28	−0.12	−0.28	−0.13	−0.43	−0.13	−0.34	−0.15
BQ	0.22	0.09	0.22	0.08	0.22	0.08	0.25	0.09	0.29	0.09
SOLR	0.08	0.05	0.07	0.04	0.07	0.04	0.10	0.03	0.07	0.03
RLRV	0.09	−0.02	0.09	−0.02	0.09	−0.02	−0.10	−0.05	−0.03	−0.04
UD	−0.12	−0.07	−0.12	−0.07	−0.12	−0.07	−0.17	−0.06	−0.12	−0.06
UT	0.14	0.06	0.57	0.24	0.38	0.10	0.20	0.06	0.34	0.33
MSL	−0.01	−0.04	−0.01	−0.04	0.05	−0.02	−0.09	−0.02	−0.01	−0.03
FA	−0.12	−0.06	−0.12	−0.06	−0.11	−0.05	0.03	−0.04	−0.14	−0.09
FTP	0.11	0.02	0.11	0.02	0.12	0.02	0.08	0.02	0.07	0.03
FTL	−0.03	0.01	0.04	0.01	0.04	0.01	−0.02	0.01	0.09	0.05
RAH	0.16	0.03	0.15	0.02	0.15	0.02	0.14	0.05	0.10	0.03
RAW	−0.06	−0.02	−0.04	−0.01	−0.05	−0.02	−0.03	−0.02	−0.43	−0.07
RTP	−0.02	−0.01	−0.02	−0.01	−0.02	−0.01	0.10	−0.01	0.01	−0.01
ANG	0.15	0.04	0.18	0.04	0.17	0.03	−0.01	0.02	0.06	0.01

^1^ ST, stature; HFE, height at the front end; TS, trunk size; CW, chest width; BD, body depth; LS, loin strength; PS, pin setting; PW, pin width; FAN, foot angle; HD, heel depth; BQ, bone quality; SORL, set of rear legs; RLRV, rear leg-rear view; UD, udder depth; UT, udder texture; MSL, median suspensory; FA, fore attachment; FTP, fore teat placement; FTL, fore teat length; RAH, rear attachment height; RAW, rear attachment width; RTP, rear teat placement; ANG, angularity; MY, milk yield; FP, milk fat percentage; PP, milk protein percentage; SCS, somatic cell score; 305MY, 305-d milk yield.

## Data Availability

The data presented in this study are available on request from the corresponding author.

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
