# Peer review of "Estimation of Genetic Parameters for Conformation Traits and Milk Production Traits in Chinese Holsteins"

_animals, 2022, doi:10.3390/ani13010100_

Round 1

Reviewer 1 Report

I evaluated the manuscript entitled “Genetic parameters analysis of conformation traits and milk 2

production traits in Chinese Holsteins” carefully. In this study, the authors estimated the genetic parameters of 23 conformation traits and 5 milk production traits of Chinese Holsteins using the DHI records of 18 cattle farms in northwest of China. These results provided a strong theoretical basis for Holstein dairy cattle breeding in Northwest China. Overall, this work is of interest and the manuscript is well prepared. I have few comments for authors’ consideration.

1. The estimated heritability of milk production traits was not mentioned in the Abstract section

2. Lines 28-35 of the Abstract describe the genetic correlation between traits in too much detail and I suggest to summarize the main results or findings of the study here.

3. Rephrasing of the sentence in lines 36-38 in the Abstract is suggested. Because the results that support this conclusion seems lack.

4. Please define the “305-d fat” and “305-d protein” in the manuscript.

5. Please define DHI in the manuscript.

6. The meaning of the expression "The final screening criteria for milk production data were 5~80 kg MY, 2000~16000kg 305MY, The sentence 2.0% to 7.0% FP and 2.0% to 5.0% PP "(lines 92-93) is not clear and needs to be revised

7. The first time an abbreviation appears in a manuscript, it should be given the full name, such as AI-REML

8. The figures legend of figure 1&2 should be revised to indicat that the upper triangle is the result of the significance test, and the lower triangle is the Pearson correlation coefficient. Also, the vertical axis of the two figures lacks the name of the traits.

9. Table 6 provide the same results with Figure 3&4. I suggest to keep one.

Reviewer 2 Report

This paper has merit and hints at the complexity of genetic selection in dairy farming.  However, some editing and additional information is required to improve the ease of reading and emphasise the practical implications the results have on genetic selection. Please see the following comments

Simple summary: This needs some work to improve readability. Please see below for examples.

Lines 10-12 could be improved to reduce the use of the word "genetic"

Line 12- Please clarify this sentence. You are talking about multiple traits and then say that one is the most important.

Line 13- "in the pasture"? what does this mean? 

Line 15- introduce the example of conformation traits where it is first mentioned in the abstract. and include stature conformation.

Line 19- Delete "It was found that"

Abstract

Line 25- This is an incomplete sentence

Line 28- Heel depth

Introduction-

Examples of how selection for conformation traits can improve cow health and production would strengthen this section.

Line 43- I don't think "role" is the correct word unless it is the actual breeding. Is it that dairy cow breeding places a high weighting on conformation traits?

Line 46-49- what is "it"?

State the objective of the study at the end of the introduction. This will help guide the reader through the reasons for the paper.

Materials and methods- a table with a brief description of each composition measure would be beneficial. This could be added to the abbreviations table 

Line 71- Do you mean farms or herds, not pastures?

Line 84- this reference is not for the technical specification.

Line 98- of less than 100 cattle? and by individuals do you mean assessors?

Line 107- farm effect?

Results

All table and figure headings- Please define abbreviations in the heading or a footer to enable the table to be standalone. In addition, more detail is needed such as that the measures are in Chinese Holstein cattle

Table 1 and 2- group the traits like how they have been grouped in table 4

Table 3- please swap min and max around

Figure 1 and 2- Missing y axis and additional information is needed in the title to make it stand alone. 

Table 6- add what bold values mean to footer

Figure 3 and 4- information in these figures is already presented in table 6 so delete or add to supplementary 

Discussion- The discussion could be developed more to relate the correlations between conformation and milk production with how it affects breeding decisions. Lines 35-39 in the abstract touched on this but it is not obvious in the discussion.  For example, selecting cows with greater capacity (i.e. trunk size and body depth) increases milk yield.

Line 265- lactation performance?
